# The Influence of Kynurenine Metabolites on Neurodegenerative Pathologies

**DOI:** 10.3390/ijms25020853

**Published:** 2024-01-10

**Authors:** Suhrud Pathak, Rishi Nadar, Shannon Kim, Keyi Liu, Manoj Govindarajulu, Preston Cook, Courtney S. Watts Alexander, Muralikrishnan Dhanasekaran, Timothy Moore

**Affiliations:** 1Department of Drug Discovery and Development, Harrison College of Pharmacy, Auburn University, Auburn, AL 36849, USA; 2Blast-Induced Neurotrauma Branch, Center for Military Psychiatry and Neuroscience, Walter Reed Army Institute of Research, Silver Spring, MD 20910, USA; 3Department of Pharmacy Practice, Harrison College of Pharmacy, Auburn University, Auburn, AL 36849, USA

**Keywords:** kynurenine pathway, neurological pathologies, inflammation, tryptophan, Alzheimer’s disease, Parkinson’s disease

## Abstract

As the kynurenine pathway’s links to inflammation, the immune system, and neurological disorders became more apparent, it attracted more and more attention. It is the main pathway through which the liver breaks down Tryptophan and the initial step in the creation of nicotinamide adenine dinucleotide (NAD+) in mammals. Immune system activation and the buildup of potentially neurotoxic substances can result from the dysregulation or overactivation of this pathway. Therefore, it is not shocking that kynurenines have been linked to neurological conditions (Depression, Parkinson’s, Alzheimer’s, Huntington’s Disease, Schizophrenia, and cognitive deficits) in relation to inflammation. Nevertheless, preclinical research has demonstrated that kynurenines are essential components of the behavioral analogs of depression and schizophrenia-like cognitive deficits in addition to mediators associated with neurological pathologies due to their neuromodulatory qualities. Neurodegenerative diseases have been extensively associated with neuroactive metabolites of the kynurenine pathway (KP) of tryptophan breakdown. In addition to being a necessary amino acid for protein synthesis, Tryptophan is also transformed into the important neurotransmitters tryptamine and serotonin in higher eukaryotes. In this article, a summary of the KP, its function in neurodegeneration, and the approaches being used currently to target the route therapeutically are discussed.

## 1. Kynurenine Pathway and Tryptophan Metabolism

The production of nicotinamide adenine dinucleotide (NAD+), which is the source of cellular energy, depends significantly on the kynurenine pathway (KP) [1]. As its relationship to inflammation, the immune system, and neurological diseases became evident, the KP intrigued more and more interest. It is the main pathway for tryptophan degradation in the liver and the beginning of NAD+ production in mammals. Tryptophan is an α-amino acid that is essential for the synthesis of proteins and is a biosynthetic precursor to various neurologically active compounds. Tryptophan research encompasses several disciplines, including both the basic sciences (biochemistry, behavioral science, immunology, pharmacology, and physiology) as well as healthcare (cardiology, diabetes, gastroenterology, hepatology), psychology (alcoholism, anxiety, depression, and schizophrenia), and neurological diseases (Alzheimer’s disease, Huntington’s Disease, and stroke). This research demonstrates the influence and adaptability of this pathway in a wide range of disorders [2]. The administration of Tryptophan has already been shown to increase oxidative stress levels and cause an imbalance between the number of free radicals present and the ability of cellular scavenging systems. The cerebral cortex is where this influence is most pronounced. It can be brought on both directly by excessive Tryptophan and indirectly by the buildup of its active metabolites. As a result, it appears that the mechanisms of brain damage seen in patients with hypertryptophanemia and neurodegenerative pathologies are influenced by abundant Tryptophan levels in the brain [3]. The most well-known bioactive chemical formed from Tryptophan is serotonin. Only a small trace of Tryptophan gets converted into serotonin, nevertheless. The production of NAD+ results from the conversion of more than 95% of Tryptophan into Kynurenine and its degradation products in the liver, involving the enzyme tryptophan dioxygenase 2, and in the cells of the immune system and in neurons, where indoleamine 2,3-dioxygenase (IDO) catalyzes the conversion of Tryptophan to kynurenines [1]. The neuroinflammatory process is significantly influenced by an imbalance in the KP metabolism [4]. The immune system gets activated, and elements with possible neurotoxic properties accumulate as a result of the dysregulation or overactivation of the KP [5]. For instance, excessive activation of the KP may cause active microglia, perivascular, and invading macrophages to produce large amounts of the excitotoxin quinolinic acid (QA). Cinnabarinic acid (CA) can act as a neuroprotective agent at low concentrations, although it can be degraded by 3-hydroxyanthranilic acid (3-HANA), a byproduct of the KP, into the well-known endogenous neurotoxins hydrogen peroxide and the free radical superoxide anion. In contrast, picolinic acid (PA) can reduce the excitatory toxicity of QA and has a dual role in immunological modulation [4].

Tryptophan is primarily metabolized by the KP (Figure 1), which produces the neurotoxic metabolites 3-hydroxykynurenine and QA and the neuroprotective metabolites kynurenic acid and PA. The IDO in human brains oxidizes Tryptophan to N-formyl-kynurenine, which then undergoes hydrolysis to produce the first stable metabolite of the KP, Kynurenine. Kynurenine is metabolized by the enzymes kynureninase, kynurenine aminotransferase, and kynurenine 3-monooxygenase to anthranilic acid (AA), kynurenic acid, or 3-hydroxykynurenine, respectively. Kynurenine is located at a branch point of the KP [6].

Due to the effects of kynurenine metabolites on excitotoxic neurotransmission, oxidative stress, neurotransmitter uptake, the modulation of neuroinflammation, amyloid aggregation, microtubule disruption, and their capacity to cause a state of dysbiosis, the KP is intricately linked to many neurodegenerative pathogenesis. Positive results from the pharmacological regulation of KP pathways suggest that it could be a feasible and explorable target for the treatment of neurodegenerative etiology [7]. In specific ROS-producing networks, Kynurenine can scavenge hydrogen peroxide and superoxide. Activated neutrophils produce less ROS when their levels of Kynurenine are higher. Kynurenine turns into Kynurenic Acid in interactions with the hydroxyl radical and peroxynitrite. In simple terms, it slows down the radicals’ effect on DNA and protein breakdown. Additionally, in rat brain homogenates, KYN can reduce the formation of ROS and lipid peroxidation brought on by pro-oxidant substances such as iron(II) sulfate, peroxynitrite, and 3-nitropropionic acid. Although Kynurenine has been used as a neuroprotective drug in numerous neurological models, the effects were thoroughly triggered by its metabolite, Kynurenic Acid [3]. Research indicates that while levels of kynurenic acid in cerebral fluid stagnate with age, they increase in blood. The findings also indicate that whereas blood and cerebrospinal fluid levels of Kynurenine and the kynurenine–tryptophan ratio are higher in older people, tryptophan levels in their blood are often lower [8].

## 2. Kynurenine Pathway and Inflammation

Many neurological, neurodegenerative, and neuropsychiatric disorders have been linked to neuroinflammation over the years. Genetics, infections, physical trauma, and psychosocial factors like stress and trauma are among the processes that can lead to neuroinflammatory conditions. Dysregulation in the kynurenine pathway (KP) of tryptophan metabolism is closely linked to these processes as a potential pathophysiological factor that could “fuel the fire” in CNS diseases [9]. Certain inflammatory stimuli cause IDO to be induced at the transcriptional level in the majority of cell types. The main IDO inducer, both in vivo and in vitro, is IFN-γ. In myeloid cells (monocyte/macrophages and dendritic cells), fibroblasts, endothelial cells, epithelial cells, smooth muscle cells, and numerous tumor cell lines, exposure to IFN-γ increases IDO transcription. To a lesser extent than IFN-γ, other inflammatory stimuli that also cause IDO include lipopolysaccharide [10], IFN-α, IFN-β, and cytotoxic T lymphocyte-associated antigen [11]. Numerous studies have demonstrated a connection between KP disregulation and microglial neuroinflammation in depression, schizophrenia, and Parkinson’s disease. A histological characteristic of neurodegenerative diseases is the abnormal buildup of protein deposits in and around neurons, including tau protein, α-synuclein, amyloid-β protein, transactive response DNA-binding protein-43, and myelin debris. In a study, in the hippocampus and prefrontal cortex in depressive individuals, there was aberrant microglial activation and increased microglial proliferation, which resulted in the production of pro-inflammatory cytokines such as IL-6, TNF-α, and interleukin-1β (IL-1β). Pro-inflammatory cytokines activate IDO, which in turn stimulates TDO, converting Tryptophan to Kynurenine. An increase in IDO activity is correlated with a corresponding increase in the conversion of available Tryptophan into Kynurenine. Pro-inflammatory cytokines also activate the QA transformation chain enzymes in microglia. A greater conversion of Kynurenine into its ultimate product, QA, is encouraged by increased activity from kynureninase (KYNU), kynurenine 3-monooxygenase (KMO), and 3-hydroxyanthranilic acid oxygenase (3-HAO). This neurotoxic metabolite increases the production of reactive oxygen species (ROS), which in turn damage surrounding tissues and cause neuroinflammation—likely the major process leading to neurodegeneration [3,12] (Figure 2). The overabundance of pro-inflammatory cytokines produced by microglia is intimately linked to KP failure. Specifically, pro-inflammatory cytokines activated microglial IDO, leading to an excess of the KP’s neurotoxic component [13]. The clinical findings indicate that in healthy women, the activation of the central kynurenine pathway increases with age, possibly due to elevated inflammation. Additionally, they found an increase in quinaldic acid with aging [14].

## 3. Kynurenine Pathway Dysfunction in Alzheimer’s Disease (AD)

An important factor contributing to neuroinflammation is an imbalance in KP metabolism [15,16,17]. Neuroinflammation and KP metabolite imbalance are interacting. Metabolic profiling was achieved in a case–control study using paired plasma and cerebrospinal fluid (CSF) samples from both clinically and biomarker-confirmed AD patients as well as cognitively healthy controls. The results indicated a specific association between amino acids and L-TRP catabolites with AD CSF biomarkers, suggesting a close relationship with core AD pathology [18]. One important amino acid that is received from diet is L-tryptophan (L-TRP). The necessary amino acid L-TRP is supplied by food. Critical downstream metabolites, including 5-hydroxytryptophan (5-HT, serotonin), melatonin (MT), and Kynurenine (KYN), are synthesized from L-TRP, a cross-kingdom starting material. In the blood, L-TRP exists in both bound and unbound forms at a ratio of roughly nine to one.

For both the central and peripheral catabolism of L-TRP, the KP is essential (Figure 1). In serum from healthy adult humans, L-TRP concentrations range from 1000 to 50,000 ng/mL. In general, KYN concentrations are only a tenth of those of L-TRP [19]. Merely 2% of L-TRP is transformed into 5-HT and MT, which are essential for controlling mood, appetite, sleep patterns, and metabolism [20]. Kynureninase (KYNU), kynurenine 3-monooxygenase (KMO), and kynurenine aminotransferase I–IV (KAT I–IV) catabolize KYN, the central metabolite of the KP, to anthranilic acid (AA), 3-hydroxy-L-kynurenine (3-HK), and kynurenic acid (KYNA), respectively. KYNU and KAT I–IV convert 3-HK to 3-hydroxyanthranilic acid (3-HANA), which is the precursor to QA, and xanthurenic acid (XA), the neurotoxic metabolite, respectively. In humans, PA is neuroprotective and produced in glial cells and neurons. A prime example of a biochemical double-edged sword is QA, which serves as a strong neurotoxin in addition to being an important metabolite [21,22,23].

Certain known factors, like molecular signals (like APOE ε4), are contentious targets for Alzheimer’s disease patients [24,25]. A multitude of pathways, including immune and inflammatory responses [26]; lipid, uric acid, and amino acid metabolism [27]; BBB dysfunction; and metal ion homeostasis, can influence Alzheimer’s disease. Since Alzheimer’s disease is a diverse neurodegenerative disease linked to aging, an accurate diagnosis that distinguishes it from normal aging is necessary. Therefore, establishing Alzheimer’s disease-related biomarkers is essential for treating, diagnosing, and preventing Alzheimer’s disease. Alzheimer’s disease patients are frequently diagnosed with serum, CSF, and imaging tests. Nonetheless, pathological portions of brain tissue and CSF are typically utilized in animal research to identify known pathological products and potential biomarkers.

In the year 2018, biomarkers were classified into three groups: A (amyloid), T (p-tau), and N (neurodegeneration, as indicated by total tau (t-tau) when appropriate) [10]. Different metabolite functions in the pathway are reflected in the differences between KP peripheral and central metabolism. In order to identify biomarkers to assist in the diagnosis of Alzheimer’s disease, serum and CSF were examined. When compared to normal controls, Alzheimer’s disease patients’ CSF contains significantly higher levels of KYNA [18,28]. Other neurodegenerative diseases like frontotemporal dementia (FTD), amyotrophic lateral sclerosis, and progressive supranuclear palsy lack increased KYNA concentrations in the CSF, unlike in patients with Alzheimer’s disease. In Alzheimer’s disease, KYNA may be notably elevated.

L-TRP has two metabolites: 5-HT and KYN. The primary byproduct of KP activation is KYN. A clinical study was conducted to evaluate Aβ, tau, and KYN in the CSF, as well as KYN, L-TRP, and 5-HT in serum, in patients with normal cognition, MCI, and Alzheimer’s disease. Because of the rise in KYN, an inflammatory signal cascade could occur during Alzheimer’s disease [29]. Because of the rise in KYN, an inflammatory signal cascade could occur during AD. Reduced functional independence and memory scores, as well as a number of inflammatory markers, were linked to higher K/T ratios. Aβ and complement systems could have been important elements that contributed to this process. Compared to controls, Alzheimer’s disease patients had reduced plasma concentrations of KYN, 3-HANA, XA, QA, and L-TRP. In particular, poor cognitive function in elderly Alzheimer’s disease patients was linked to elevated QA. According to another study, peripheral KP activation in Alzheimer’s disease was demonstrated by the decrease in plasma concentrations of L-TRP and KYNA and the increase in QA [30].

In Alzheimer’s disease patients, there was an inverse correlation between CSF p-tau and t-tau and plasma KYN and PA. Furthermore, in the CSF of Alzheimer’s disease patients, elevated 3-HK/KYN ratios correlated with t-tau [31]. Moreover, the CSF of Alzheimer’s disease patients exhibited noticeably greater concentrations of the L-TRP catabolites QA and KYNA. These increased concentrations showed correlations with tau and p-tau-181 or with CSF Aβ1-42, similar to what other L-TRP pathway intermediates accomplished. Patients with Alzheimer’s disease had lower concentrations of KYNA in their CSF. Additionally, there was a negative correlation found between the severity of Alzheimer’s disease and serum KYN and QA levels, as well as a strong correlation between the two variables’ contents in the CSF. It has been proposed that rather than QA levels alone, changes in the ratio of KYNA to QA are connected with the development of inflammatory-mediated neuropathology [32]. In both Alzheimer’s disease patients and control subjects, KYN metabolites accumulated in serum and CSF in a manner consistent with aging. In contrast, Alzheimer’s disease patients’ CSF had a marked decrease in KYNA. Reduced neurogenesis and increased excitotoxicity in neurodegenerative diseases may be caused by age- and disease-specific alterations in cerebral KP activity [33]. These results validate the role of the KP in Alzheimer’s disease pathogenesis and the idea of therapeutic KP regulation in Alzheimer’s disease treatment. Moreover, it was discovered that the severity of Alzheimer’s disease was correlated with the following tests: 3-HANA (plasma), XA (plasma), QA (plasma and CSF), L-TRP (plasma, serum, and urine), and KYN (plasma, serum, and CSF). KP metabolites such as QA and KYN might be helpful biomarkers for the early diagnosis of Alzheimer’s disease.

The KP’s metabolism of L-TRP controls tumor growth, immunity, and inflammation. Numerous metabolic variables, including inflammatory ones, alter KP signaling. Therefore, early identification of metabolites can aid in the early diagnosis and treatment of AD. Research findings indicate that peripheral inflammatory factors were crucial in the initial development of the blood–cerebrospinal fluid barrier [34,35]. Since peripheral inflammation is linked to neuroinflammation, we should approach inflammation systemically. Second, the majority of the intricate molecular mechanisms underlying KP metabolites are yet unknown. When considering molecular pathways and the state of biomarker exploration, distinct biomarkers are the most suitable. The anti-neuroinflammatory properties of KYN, 3-HANA, and CA help prevent Alzheimer’s disease. It has been discovered that KYNA may cause cognitive impairment. The neurotoxicity of 3-HK, QA, and PA facilitates Alzheimer’s disease. Additionally, QA has been suggested to cause neuroinflammation and have a close relationship with the production of p-tau and Aβ [15,36,37].

Based on current research, increased QA in Alzheimer’s disease leads to pathogenic alterations and neuroinflammation. However, in comparable investigations, KYN may be a more useful biomarker. This could be a result of KYN’s ability to traverse the BBB via the large neutral amino acid transporter, whereas QA mostly diffuses weakly over the BBB and is found in the central nervous system [4]. Researchers suggest that QA may be a better therapeutic target than KYN while KYN may be more useful as a biomarker. Subsequent research endeavors should meticulously examine the biological roles of diverse significant KP metabolites. In order to connect KP metabolites with AD, more mechanistic research is required. Nonetheless, it is probably advantageous to alter the KP at different locations along the pathway. Future treatment approaches for neurological disorders include neutralizing antibodies, enzyme inhibitors, and KP metabolites [4].

## 4. Kynurenine Pathway Dysfunction in Parkinson’s Disease (PD)

The KP of tryptophan catabolism is a key regulator of the immune response and is believed to be involved in inflammatory and neurotoxic processes in Parkinson’s disease. The KP produces a number of neuroactive chemicals that can be neurotoxic, neuroprotective, or immunomodulatory. KYNA, one of these metabolites generated by astrocytes, is thought to be neuroprotective. Some KP metabolites may be employed as prognostic biomarkers, and the use of pharmacological modulators of the KP enzymes may constitute a potential treatment method for Parkinson’s disease [38]. The KP is one of the key immune-response-regulating systems. The induction of the KP regulating enzyme IDO-1, KP activation, and TRP depletion are all implicated in the establishment of pregnancy-related immunological tolerance and tumor tenacity. IDO-1 activation in dendritic cells totally inhibits T-cell clonal growth [39].

The cellular manifestation of KP in the brain is currently poorly known. While cellular KP enzymes are completely expressed in monocytic cells such as macrophages and microglia [40], we discovered that the KP is only partially expressed (i.e., some particular enzymes of the route are not produced) in human astrocytes, neurons, oligodendrocytes, and endothelial cells. Neurons and astrocytes have neuroprotective KP profiles whereas activated microglia and invading macrophages are neurotoxic. Different KP compounds can be neurotoxic, neuroprotective, or immunomodulatory [41]. Among these, excitotoxin quinolinic acid (QUIN) appears to be the most significant, causing severe neuronal cell death and persistent malfunction via at least nine distinct pathways.

The KP is only activated by IDO-1 during a neuroinflammatory state, which can be acute, low-grade, or progressive. IDO-1 is known to be activated by a variety of inflammatory mediators, including IFN-γ, TNF-α, TLRs 1–6 and 9, Lipopolysaccharides [10], amyloid, and viral proteins. IDO-1 is the primary TRP catabolic enzyme in the brain [42]. The KP is known to have a role in a variety of neuroinflammatory illnesses such as Alzheimer’s disease, Huntington’s Disease, amyotrophic lateral sclerosis, multiple sclerosis, infections, brain tumors, brain trauma, and neuropsychiatric disorders such as depression, suicide, schizophrenia, and autism [42]. The first description of KP impairment in Parkinson’s disease was published in the early 1990s. TRP/KYN and KYNA/TRP ratios were considerably elevated in the frontal cortex, putamen, and putamen and pars compacta of the SN (SNpc) in Parkinson’s disease patients, according to Ogawa et al., while 3-HK levels were greater in the putamen and SNpc.

KYN and KYNA concentrations in the frontal brain were dramatically lowered in Parkinson’s disease patients who took L-dopa. Similarly, in L-dopa-treated Parkinson’s disease patients, the KYN/3-HK ratio was considerably reduced in the frontal cortex and SNpc but only in the putamen. Another group discovered a drop in KYNA levels in the cortical areas, caudate, and cerebellum in Parkinson’s disease patients [43]. The decreased capacity to prevent excitotoxicity via the NMDA receptors, caused by QUIN and/or glutamate excess, is a direct result of this reduced level of endogenous KYNA [44]. In the SNpc in MPTP-treated animals, the expression of kynurenine aminotransferase-I (KAT-I), the KP enzyme that leads to KYNA production, is reduced. The same group discovered that KAT-I and tyrosine hydroxylase (TH) are co-expressed in the same SNpc neurons and that 6-OHDA injection into the lateral ventricle of adult rats resulted in the death of most nigral neurons.

The investigators also discovered that astrocytes in the SNpc express KAT-I under normal settings and increase after 6-OHDA administration, but microglia only became immunoreactive to KAT-I after 6-OHDA administration. KYN/TRP ratios have also been shown to be higher in the CSF and serum in Parkinson’s disease patients when compared to age- and gender-matched healthy controls [45,46]. According to evidence from post-mortem PD brain tissue and MPTP-treated animals, KAT-I and KAT-II activity were dramatically decreased, which corresponded to lower plasma KYNA levels [47]. There have also been reports of KP alterations in peripheral organs in Parkinson’s disease. These studies offer convincing proof that TRP catabolism is shifting in favor of 3-HK and QUIN, which lowers KYNA concentrations and causes neurotoxicity and cell death.

In the past ten years, psychoneuroimmunology and neurodegeneration in a variety of neurological conditions—Parkinson’s disease, specifically—have been directly linked by metabolomics of the KP [48,49]. This holds significant importance in comprehending the interconnected structure between neuroinflammation and neurodegeneration, wherein innate immunity plays a pivotal role in pathobiology and the advancement of illness. It is now well documented that aged individuals with impaired TRP and tyrosine metabolism are also associated with chronic low-grade inflammation, which concurrently activates the KP [50]. Protein synthesis, KP metabolism, and the well-established serotonin (5-HT)/melatonin pathway all depend on TRP, whose regulation is strictly regulated. Nevertheless, this equilibrium may be upset by immunological activation, particularly in cases of persistent low-grade inflammation.

Translational validity in the development of Parkinson’s disease has been proven by immune dysregulation and poor neurological function. In fact, a pathogenetic connection has been shown between immunological activation and neuropsychiatric conditions such as Alzheimer’s disease, Parkinson’s disease, anxiety, and schizophrenia [51].

The elevation of significant pro-inflammatory markers like IL-6, CRP, and MCP-1 in the CSF of Parkinson’s disease patients was initially demonstrated by Lindqvist D (2012). These increases may trigger the emergence of non-motor symptoms in Parkinson’s disease (PD), which are linked to non-motor symptoms such as fatigue, depression, and cognitive impairment [52]. This phenomenon is intimately linked to illness behavior that is brought on by protracted-immunity-induced elevated KP activation. Thus, it is quite likely that PD also involves IFN-γ-mediated activation of IDO-1, which can be further enhanced by other cytokines like IL-1β and toll-like receptor agonists like LPS. Furthermore, it was shown by Widner et al. (2002a and 2002b) [45,46] that in both the peripheral and central nervous systems (CRS) in Parkinson’s disease (PD) patients, elevated TRP catabolism, as indicated by the KYN/TRP ratio, positively corresponds to the inflammatory marker neopterin. This offers more proof that immunological mediators support the elevation of KP-activation in Parkinson’s disease [46].

## 5. Kynurenine Pathway Dysfunction in Other Neurodegenerative Diseases

The metabolites of KP have defined potential relationships with various neurological disorders based on their effects on several neurotransmitter pathways. QA functions as an NMDA receptor agonist, leading to synaptic excitotoxicity [53]. The synaptic dysfunction caused by QA is not only implicated in AD and PD but is also considered a contributing factor to other cognitive impairment conditions [54]. KYNA significantly acts as a recognized NMDA receptor antagonist and performs a neuroprotective effect in the CNS. Research has also suggested that KYNA results in a downregulation of the dopaminergic neuron system activity [55,56]; an inhibitory effect when it comes to α7nAChR, the Nicotinic receptor in the CNS [57,58]; and negative impacts on CNS GABAergic activity [59]. Except for the mentioned neurodegeneration diseases, various pieces of research report a connection with KP, including Huntington’s Disease (HD), Lewy body Dementia (LBD), Amyotrophic lateral sclerosis (ALS), frontotemporal dementia (FTD), and multiple system atrophy (MSA).

Huntington’s Disease (HD) is a single-gene genetic disease that suggests GABAergic activity degeneration as the major hypothesis for its pathology. Although there is a lack of evidence for PK getting involved in HD processes [60], downstream metabolites of KP, including QA, 3-HK, and QUIN, exhibit identical biological activity to HD. Typically, QA-induced excitotoxicity and neuroinflammation were noted in QA-lesioned rodents applied as the major experimental model for HD before the causative gene was identified [61].

With the development of the theoretical study of HD pathology, scientists found a significant difference in the 3-HK level in the CSF [62], whereas the QA shows a stable level as compared to the control group. An increased level of 3-HK was demonstrated in the striatum and the cortex in HD and was implicated in neuron loss as they prompted apoptosis [62]. The enzyme kynurenine 3-monooxygenase (KMO) is the critical enzyme that mediates the transamination from KA to 3-HK, taking QUIN as a final metabolite. In contrast, the kynurenine aminotransferase (KAT) enzymes catalyze the irreversible transamination from KA to KYNA. Scientists suggest that inhibiting KMO could aid HD patients in lowering 3-HK and QUIN levels in neurons, promoting the conversion of KA into its neuroprotective metabolite [63].

Lewy body Dementia (LBD) is another major cause of dementia. The abnormal aggregate of the alpha-synuclein protein caused an accumulation of Lewy body deposits and induced the pathology change in cognitive ability in one study [64]. The level of Kynurenine (KYN) and Kynurenic acid (KA) was believed to have a similar effect on LBD patients with Alzheimer’s disease. A study conducted by Solvang SH in 2019 collected five-year data related to the kynurenine pathway (KP) from 65 LBD patients. The study suggested a non-linear relationship between the ratios between KYN and TRP and between KA and KYN and the Mini-Mental State Examination (MMSE) result. Neither high nor low levels of IDO-generated KYN exhibited an excellent performance when it came to the MMSE. Higher levels of KKR indicated a significant association with increasing neuropsychiatric symptoms, especially hallucinations [65]. Additionally, LBD is demonstrated to have a considerable association with the apolipoprotein E ε4 gene variant (APOEε4). APOE expression is suppressed by pro-inflammatory cytokines in monocytes, which affects the activation of the kynurenine pathway [66,67,68]. There have been no differences between the KYN, KA, and QA levels observed in patients with different alleles of APOE. However, the absence of the ε4 allele indicates a higher possibility of cognition problems induced by the disordered kynurenine-pathway-related chemical level [66].

Amyotrophic lateral sclerosis (ALS) and frontotemporal dementia (FTD) are both early-onset neurodegenerative disorders with a risk of death. Although there is a considerable difference in clinical performance between ALS and FTD, more and more evidence shows a mixed disease phenotype and genomic changes are present in both [69]. However, there is a lack of further understanding to distinguish early onset neurodegenerative disorders from other diseases and offer precise diagnosis and treatment. At present, the diagnosis of FTD is highly dependent on the presence of behavioral and cognitive symptoms in combination with neuroimaging without the assistance of liquid biomarkers. On the other hand, El Escorial criteria specified the diagnostic criteria from the perspective of movement disorders for ALS [70]. The current obstacle of ALS and FTD has inspired scientists to start following the KP that induces the neuroinflammation response as a novel strategy for investigation [71]. The consistency of the biochemical phenomena caused by the neurotoxicity of QUIN and the pathology of ALS and FTD have suggested a possible causal relationship. The hypotheses related to the pathogenesis of the two diseases include QUIN-inducing neuroinflammation, QUIN-inducing iNOS synthesis, QUIN-inducing NMDAR activation and Glutamate Excitotoxicity, and QUIN-inducing mitochondrial dysfunction and ATP death among others [72].

Overall, multiple mechanisms of the KP have given its metabolisms (KPM) very potential as biological markers in clinical settings, especially for ALS. KPM is a group of small molecules within a stable state and widely distributed in the CSF, blood, and urine. Unlike TRP and KYN, which has a higher permeability through the blood–brain barrier (BBB), QUIN and KA are produced locally in situ from Kynurenine, and the higher level of QUIN and KA in the CSF reflects a higher neuroinflammation response [73]. Research conducted by Alarcan H in 2021 recruited 42 ALS patients and forty controllers to study the early disorder of KP in the CSF. KYNA, KYN, and QUIN were expected to be most relevant, with a tendency to decrease in ALS subjects, and in contrast, the 3-HK/KYNA and QUIN/KYNA ratios tended to be higher in ALS subjects [73]. The observed variations in KPM also suggest a potential association with age or courses of disease for ALS [65]. However, the neuroinflammation response induced by the KP commonly occurs in several neuron diseases. Thus, the KPM assessment is more valuable when applied as a prognostic/progression marker and differentiates the disease subtypes instead of using them for diagnosis.

Multiple system atrophy is a neurological disorder primarily symptomized by the loss of neurons and the presence of overabundant or enlarged glial cells within multiple locations of a patient’s central nervous system. Although strongly indicated by the histopathological hallmark that comprises oligodendroglial cytoplasmic inclusions, which contain accumulated, improperly folded α-synuclein, a definitive clinical diagnosis for the condition can only be reached through the presence of autonomic dysfunction and Parkinsonism in the patient [74]. The mechanisms involved in the kynurenine pathway have been implicated in a number of neurodegenerative conditions. Kynurenines, which are, as previously elucidated, metabolites produced within the kynurenine pathway, play a significant role in the development of chronic inflammation. Kynurenines negatively impact glutamatergic neurotransmission and exhibit direct toxic effects on neurons and glial cells, both of which are known to be damaged in the pathophysiology of multiple sclerosis. Furthermore, another intermediate metabolite of the kynurenine pathway, quinolinic acid, is considered an established neurotoxic agent [75].

Multiple sclerosis (MS) is a complex and debilitating autoimmune disease that affects the central nervous system, leading to a wide range of neurological symptoms. Physiological symptoms include the degeneration of myelin sheaths surrounding neurons, chronic inflammation of neural tissues, and formation of CNS lesions, leading to various neurological symptoms [76]. The pathogenesis is not entirely understood, but genetic and environmental elements contribute to disease presentation. After the first enzymatic rate-limiting step, the KP pathway metabolizes Tryptophan in three distinct branches. IDO catalyzes this first step. Research has supported that the link between the KP and the immune system lies in IDO activity since IDO appears to have some connection to anti-inflammatory properties and the effect on T helper and regulatory cells when considering its activation by cytokines [5]. KP metabolites are directly toxic to glia and neurons [77]. The activation of the KP pathway results in changing levels of kynurenine metabolites, and elevated levels of these metabolites have been found in the cerebrospinal fluid (CSF) of patients presenting with all types of MS [75]. When it comes to understanding the intricate connections between types of multiple sclerosis, the kynurenine pathway provides valuable insights into the underlying mechanisms of the disease, offering potential avenues for therapeutic interventions and the development of targeted treatments to alleviate the impact of MS on affected individuals.

## 6. Therapeutic Targets Modulate the Kynurenine Pathway

Due to the sheer variety of neurodegenerative diseases that can develop in the body, there is an immense range of pathophysiological factors behind the symptoms associated with neurodegeneration. This, in turn, means that there are a great many therapeutic targets in the body that can be altered to modulate the severity of these neurodegenerative conditions. A therapeutic target is defined as a biochemical factor—typically a molecule, pathway, or physiological reaction—that is affected by a specific therapeutic drug, resulting in the beneficial effects associated with that drug [78,79]. One metabolic process that has been implicated heavily in the mechanisms of many neurodegenerative disorders is the kynurenine pathway, which functions primarily as the body’s means to degrade the essential amino acid tryptophan [80]. The pathway’s considerable involvement in disease pathology means that it is also a viable target for various therapeutic processes.

It has been noted in many studies that kynurenic acid, one of the many byproducts of the kynurenine pathway, displays an increased concentration in the brain tissue or the cerebrospinal fluid of individuals with schizophrenia—a condition symptomized almost entirely by its reduction of cognitive function in patients. This increase has been observed to be several-fold and especially prominent within the prefrontal cortex in diseased individuals, leading to the development of hypo-frontality in that area of their brains. Hypo-frontality, defined as a slowing of EEG activity localized frontally [81], coincides with decreased metabolic function and is one of the indicators of schizophrenia along with various other neurological conditions. It has been hypothesized that a reduction in the quantity and activity of the enzymes kynurenine 3-monooxygenase (KMO) and 3-hydroxy-anthranilic acid oxidase (3HAO), along with a reduction in the quantity and activity of their corresponding mRNA, is the reason for the association between increased levels of kynurenic acid and schizophrenia. Furthermore, a correlation between levels of kynurenic acid in the cerebrospinal fluid and the KMO single nucleotide polymorphism rs1053230 has been documented, supporting the notion that greater amounts of kynurenic acid production can be caused by simple genetic alternations. Finally, kynurenic acid is known to decrease extracellular glutamate levels, leading to the observation of impaired cognitive function in rat models [82]. One of the clinical studies on the vagus nerve stimulation (VNS) mechanism of action in recurrent epileptic seizures showed that the neuroprotective and anticonvulsant tryptophan metabolite anthranilic acid is consistently elevated in response to vagus nerve stimulation. Furthermore, a decrease in the frequency of seizures and an improvement in mood are linked to elevated anthranilic acid levels [83]. An analysis of clinical studies involving patients diagnosed with Alzheimer’s-type dementia indicated that the kynurenine pathway is impaired in Alzheimer’s-type dementia patients. The study also found that after citalopram treatment, there was a significant correlation between the levels of Kynurenine and Tryptophan for Alzheimer’s-type dementia patients and an increase in serotonin (5-HT)/Tryptophan [84]. Acute tryptophan depletion significantly reduces cognitive function in patients with Alzheimer-type dementia, according to a study by R. J. Porter [85].

The KMO enzyme has also been directly and considerably implicated in the pathology of neurodegenerative diseases. One study of mice models observed that the targeted deletion of KMO-related genes, which had the effect of preventing KMO production, resulted in a significant decrease in the enzymatic product of KMO, 3-hydroxykynurenine (3-HK) [86]. This is significant as 3-HK is considered a neurotoxin due to both its generation of free radicals as well as its functionality as an agonist for N-methyl-d-aspartic acid [87]. 3-HK exhibits particularly toxic effects on cortical and striatal neurons [88] and is also known to prevent cytokine-induced apoptosis, a mechanism of cellular regulation, within them [87]. One method by which to alleviate the effects of 3-HK is inhibiting its uptake into the cell, as uptake has been shown to be a critical factor in its toxicity [88]. Huntington’s Disease is a primary example of a neurodegenerative disorder with a pathology that correlates to the action of KMO. Brain regions with the strongest associations to HD-related neurodegeneration have been observed to contain aggregations of reactive microglia [87], which are the central nervous system’s immune cells involved in processes such as brain homeostasis and the development of neurological disorders [89]. Upon induction from pro-inflammatory stimuli, microglia have been known to express KMO, which leads to their production of quinolinic acid [87], another neurotoxin known to play a large role in neurodegenerative disease pathology. While typically present in the human brain and cerebrospinal fluid in only nanomolar concentrations, abnormally increased amounts of quinolone acid may have considerable consequences. Like 3-HK, quinolone acid is an N-methyl-D-aspartic acid receptor, but it can also exhibit effects on transcription factors, cell death, presynaptic receptors, oxidative stress, and energy dysfunction [17]. Notably, in patients with Huntington’s Disease, the central nervous system has been known to favor the KMO branch of the kynurenine pathway. This shift eventually results in the accumulation of quinolinic acid and the occurrence of the toxic effects that it can have in elevated amounts. Further supporting the association of KMO in neurodegenerative diseases is the fact that the KMO inhibitor drug, JM6, has been noted for its ability to inhibit the loss of synapses, lower microglial activation, and increase life span in an HD mouse model [87]. The clinical trial showed that the Indole-3-propionic acid (IPA), which is formed by the gut microbiota, helps prevent damage to microglia from inflammation and promotes neuronal function. *Peptostreptococcus anaerobius* and *Clostridium sporogenes* are the only bacteria that can synthesize IPA from dietary Tryptophan. An animal model of steatohepatitis has demonstrated the anti-inflammatory properties of circulating IPA, and it has also been demonstrated that it mitigates ischemia-induced cerebral lesions by lowering lipid peroxidation and DNA damage [90]. Oats, bananas, dried prunes, milk, tuna fish, cheese, bread, chicken, turkey, peanuts, and chocolate are a few foods that often contain Tryptophan [91].

## 7. Conclusions

Evaluating the precise mechanism behind a neurological condition is challenging yet crucial due to the intricacy of neurodegenerative disorders. Because neurological pathologies are associated with increased morbidity and mortality, clinical research in this field is conducted on animals rather than human beings. Clinical trials on humans are nonetheless required to fully understand the impact of therapeutic targets and how they might help mitigate disease. An increasing amount of research links the pathophysiology of several neurocognitive and neurodegenerative pathologies to metabolites of the kynurenine pathway. In order to support earlier metabolic research in patients, new genetic and pharmacological methods in model organisms have been used to imply that normalizing circulation via the KP may be a viable treatment strategy for various pathologies.

## Figures and Tables

**Figure 1 ijms-25-00853-f001:**
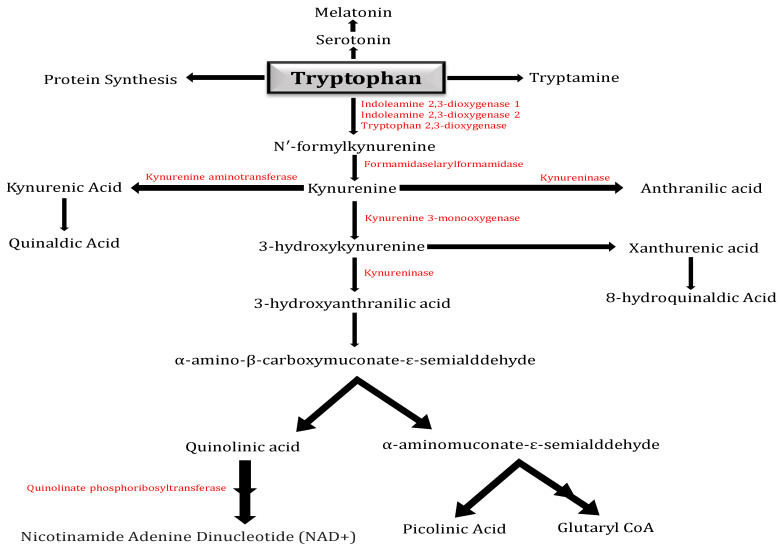
Kynurenine pathway: Diagram illustrating the kynurenine pathway (KP), which in higher eukaryotes is responsible for approximately 95% of tryptophan metabolism. The pathway has a crucial branch point in the context of neurodegeneration at the metabolism of Kynurenine. Kynurenine 3-monooxygenase (KMO) converts Kynurenine into the neurotoxic free-radical generator 3-hydroxy-L-kynurenine (3-HK), which then advances to the synthesis of the neurotoxic metabolites quinolinic acid (QA) and 3-hydroxyanthranilic acid (3-HANA). The kynurenine aminotransferase (KAT) family of enzymes can additionally metabolize Kynurenine, resulting in kynurenic acid (KYNA), which has neuroprotective properties.

**Figure 2 ijms-25-00853-f002:**
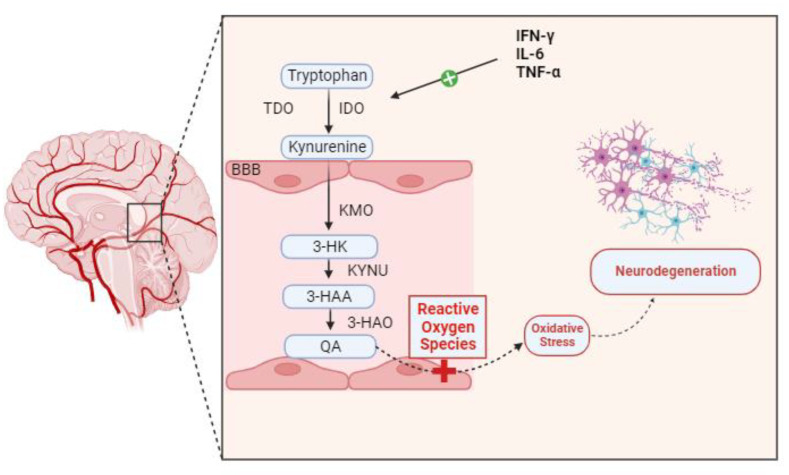
Pro-inflammatory cytokines, including TNF-α, IFN-γ, and IL-6, activate IDO, which in consequence activates TDO, converting Tryptophan to Kynurenine. The conversion of accessible Tryptophan into Kynurenine increases proportionately with increased IDO activation. Pro-inflammatory cytokines also stimulate the activity of the microglia’s QA transformation chain enzymes. Increased activity from Kynureninase (KYNU), kynurenine 3-monooxygenase (KMO), and 3-hydroxyanthranilic acid oxygenase (3-HAO) encourages a higher conversion of Kynurenine into its final product, QA. This neurotoxic metabolite can enhance oxidative stress by favorably influencing the generation of ROS. Neuroinflammation and oxidative stress result in neurodegeneration.

## Data Availability

Not applicable.

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
