# Peer review of "The Influence of Kynurenine Metabolites on Neurodegenerative Pathologies"

_ijms, 2024, doi:10.3390/ijms25020853_

Round 1
Reviewer 1 Report
Comments and Suggestions for Authors
Summary of the Manuscript:
Reviewed the manuscript "The Influence of Kynurenine Metabolites on Neurodegenerative Pathologies" by Suhrud Pathak et al. This manuscript delves into the kynurenine pathway (KP) and its implications in neurodegenerative diseases. It highlights the pivotal role of the KP in tryptophan degradation and its relation to the immune system and inflammation. The paper discusses how dysregulation of this pathway can lead to the accumulation of neurotoxic substances and influence neurological conditions like depression, schizophrenia, and Alzheimer's disease, Parkinson's disease, multiple sclerosis, and others. It also explores the therapeutic potential of targeting the KP in treating these diseases, especially considering the pathway's role in neurotoxicity and neuroprotection​.
Strengths:
- Comprehensive Coverage: The paper extensively covers the role of the KP in various neurodegenerative diseases, providing a thorough overview of the current understanding of the pathway.
- Depth of Research: The inclusion of detailed mechanisms by which KP metabolites influence disease pathology is a significant strength.
- Therapeutic Relevance: The discussion of potential therapeutic targets within the KP provides valuable insights into future treatment strategies.
Weaknesses:
- Unclear Biochemical Explanations: The paper does not adequately explain the specific biochemical mechanisms at play within the KP, leading to potential confusion or misinterpretation of the pathway's role in neurodegenerative diseases.
- Figure legend is overly simplistic and fails to provide a detailed explanation of the figure's content, missing an opportunity to enhance reader comprehension.
Recommendations:
- Enhance Figure 1: Redesign Figure 1 with higher resolution and improved font clarity. Expand the legend to provide a comprehensive explanation of the KP's involvement in neurodegenerative diseases. This should include a detailed description of the pathway's components and their specific roles in disease pathogenesis.
- Incorporate Additional Visuals: Introduce more figures and tables that depict various aspects of the KP, such as its metabolic process, interaction with neurodegenerative diseases, and potential therapeutic targets. These visuals should be designed to aid in understanding complex biochemical processes and research findings.
- Focus on Clinical Relevance: Augment the content with more information on human clinical studies. This could involve discussing ongoing or completed trials that examine the therapeutic potential of targeting the KP in human patients, offering a more holistic view of the research.
- Detail Specific Biochemical Mechanisms: Provide a clearer and more detailed explanation of the biochemical mechanisms by which KP metabolites influence neurodegenerative diseases. This should include a discussion of the molecular pathways involved and how they contribute to disease pathology.
Overall Recommendation: This manuscript requires a major revision to address the issues with the visual aids, expand its clinical perspective, and improve the clarity of the biochemical mechanisms involved. Once these revisions are made, the paper could offer significant contributions to the field of neurodegenerative disease research.
Reviewer 2 Report
Comments and Suggestions for Authors
This review indicates the kynurenine metabolites as potential indicators of the development of neurological disorders. The manuscript is well-written and the subject is very important due to the lack of effective treatment against many neurodegenerative disorders including Alzheimer’s and Parkinson’s diseases. The manuscript is presented in a well-structured manner, as well as the conclusions are consistent and based on many data cited in the publication. There are only a few comments/questions below:
11. Page 1, line 19: Specify what kind of neurological conditions the authors consider?
22. Add in a few sentences what is the exogenous source of tryptophan, and also whether there are any consequences of deficiency or excess of this amnio acid?
33. Are there any changes in KP metabolites depending on age or gender?
44. I suggest supplementing the work with a figure showing the relationships of the KP with neurotoxic or neuroprotective processes occurring in the brain during neuropathology.
55. Page 10, line 440: Please, add to the manuscript some information about the latest and most interesting clinical trials and advances in targeting kynurenine pathway during neurodegenerative disorders.
In addition to the above comments, I think the work is great and provides a huge overview of the knowledge on the subject, that leads me to accept the manuscript only after minor revision.
Round 2
Reviewer 1 Report
Comments and Suggestions for Authors
Upon reviewing the revised manuscript "The Influence of Kynurenine Metabolites on Neurodegenerative Pathologies" by Suhrud Pathak et al., I am pleased to note that the authors have effectively addressed all previous concerns. The enhancements in visual aids, detailed biochemical explanations, and incorporation of clinical relevance significantly enrich the manuscript. The paper now provides a comprehensive, clear, and insightful exploration of the kynurenine pathway in neurodegenerative diseases. I commend the authors for their diligent revisions and recommend the manuscript for publication in its current form, as it makes a valuable contribution to the field.